# How dentists and oral and maxillofacial surgeons deal with tooth extraction without a valid clinical indication

Dyonne Liesbeth Maria Broers[1]*, Leander Dubois[1,2,3], Jan de Lange[1,2,4], Jos Victor Marie Welie[5,6], Wolter Gerrit Brands[6,7], Maria Barbara Diana Lagas[1], Jan Joseph Mathieu Bruers[1,7], Ad de Jongh[1,8,9,10,11]

1 Academic Centre for Dentistry Amsterdam, Amsterdam, University of Amsterdam and Vrije Universiteit, Amsterdam, The Netherlands, 2 Department of Oral and Maxillofacial Surgery, Amsterdam University Medical Centers, University of Amsterdam, Amsterdam, The Netherlands, 3 Department of Oral and Maxillofacial Surgery, St Antonius Hospital, Nieuwegein/Utrecht/Woerden, The Netherlands, 4 Department of Oral and Maxillofacial Surgery, Isala Klinieken Zwolle, Zwolle, The Netherlands, 5 University College Maastricht, Maastricht University, Maastricht, The Netherlands, 6 St. André International Center for Ethics and Integrity, Saint-André-d'Olérargues, France, 7 Royal Dutch Dental Association (KNMT), Utrecht, The Netherlands, 8 Research department PSYTREC, Bilthoven, The Netherlands, 9 School of Health Sciences, Salford University, Manchester, United Kingdom, 10 Institute of Health and Society, University of Worcester, Worcester, United Kingdom, 11 School of Psychology, Queen's University Belfast, Belfast, Northern Ireland

* d.l.m.broers@acta.nl

**Data Availability Statement:** The data collection was commissioned by the Royal Dutch Dental Association (KNMT) and carried out by an independent research agency ('Third Party

## Abstract

### Objectives

This study pertains to a secondary data analysis aimed at determining differences between oral and maxillofacial surgeons (OMFSs) and dentists handling dental extractions without an evident clinical indication.

### Study design

A survey of 18 questions was conducted among 256 OMFSs in the Netherlands and a random sample of 800 dentists Respondents could answer the questions in writing or online. The data was collected in the period from November 2019 to January 2020, during which two reminders were sent. Analysis of the data took place via descriptive statistics and Chi Square test.

### Results

The response rate was 28.1% (n = 72) for OMFSs and 30.3% (n = 242) for dentists. In the past three years, 81.9% (n = 59) of the OMFSs and 68.0% (n = 164) of the dentists received a request for extraction without a clinical indication. The most common reasons were financial and severe dental fear (OMFSs: 64.9 and 50.9% vs dentists: 77.4 and 36.5%). Dentists were significantly more likely (75.6%, n = 114) than OMFS (60.7%, n = 34) to comply with their last extraction request without a clinical indication. Almost none of them regretted the extraction afterwards. As for the request itself, it was found that 17.5% (n = 10) of the OMFSs and 12.5% (n = 20) of the dentists did not check for patients' mental competency (p = 0.352).

Research Institute'). It was determined that the KNMT remained the owner of the collected coded data. The coded data supporting the findings of this study are available and can be requested from the KNMT, via the Research Department (staatvandemondzorg@knmt.nl). Furthermore, it can be confirmed that the authors did not receive any special privileges in accessing the data that other researchers would not have.

**Funding:** The authors received no specific funding for this work.

**Competing interests:** The authors whose names are listed certify that they have no affiliations with or involvement in any organization or entity with any financial interest (such as honoraria; educational grants; participation in speakers' bureaus; membership, employment, consultancies, stock ownership, or other equity interest; and expert testimony or patent-licensing arrangements), or non-financial interest (such as personal or professional relationships, affiliations, knowledge or beliefs) in the subject matter or materials discussed in this manuscript.

## Conclusions

Given that most of the interviewed dental professionals complied with non-dental extraction requests when such extractions are ethically and legally precarious, recommendations for handling such requests are greatly needed.

## Introduction

Oral and maxillofacial surgeons (OMFSs) and dentists often deal with a request for a dental extraction. There are good indications, grounded in dental science, to remove permanent teeth [1]. Most often, extraction is recommended by the dentist or the OMFS because of advanced caries or periodontitis [2–5]. But sometimes, the patient wishes to have a tooth removed, even though the dentist or OMFS does not think such extraction is indicated. Of course, there are grey zones between both types of extractions.

An extraction request without a clinical reason can be related to financial, or cultural issues. Lack of financial resources is the most common reason for such a request [6]. Furthermore, there is a tradition of extracting follicles or incisors in certain cultures (e.g., among the Dinka and the Nuer from Sudan) [7, 8]. Also, a severe mental health condition can lead to such a request, for instance, dental phobia, post-traumatic stress disorder (PTSD), somatoform pain disorder, or a mental health condition involving a disturbed body awareness such as body dysmorphic disorder (BDD) and body integrity identity disorder (BIID) [9–11]. In the case of dental phobia, some patients believe that by extracting teeth, they prevent further frightening and painful dental treatments.

It has been estimated that 3.6 to 5.9% of the extractions are conducted without a clinical indication [12]. This is in line with a study that investigated how often patients request an extraction while the dental professional refuses to carry it out [6]. In a sample of 800 Dutch dentists, we showed that 6% of the dentists reported being confronted with a request for extraction without a clinical indication in the last three years. Financial reasons were most often reported (about 50%), followed by psychological and financial reasons combined (28%), psychological reasons only (18%), and other reasons (4%). Of all extraction requests without a clinical indication, 76% appeared to be granted. The researchers concluded that requests for such extractions are relatively common and that "while dentists are reluctant, in theory, they are likely to grant such a request in everyday practice, particularly if the patient cannot afford an indicated restorative treatment" [6]. These findings are striking, given that removing one or more permanent teeth is an irreversible act that can cause damage to the surrounding oral tissues. To do so without a sound clinical reason violates the ethical principle of non-maleficence (don't cause harm), as enshrined in most international and national codes of (dental) ethics [13]. By the same token, it also violates the legal standard of care as adopted in The Netherlands and many other jurisdictions [14–16].

Both OMFSs and dentists perform teeth extractions. Given their different education and occupational contexts, the question arises as to whether OMFSs and dentists also respond differently to requests for extractions that have no clear clinical indication. For example, are OMFSs (as a result of the medical training) more inclined to assess the decision-making competence of patients who make such requests? Because research on this topic is lacking, we decided to further analyze our previously collected data [6] to investigate possible differences between OMFSs and dentists. We sought to determine 1) how often OMFSs and dentists encounter an extraction request without a clinical indication, 2) the reasons for such an extraction request, 3) how likely it was that such a request was granted, 4) how they evaluated their

own decision to grant or not to grant the request, and 5) the extent to which they checked patients' competency.

## Materials and methods

### Design and participants

For this survey, questionnaires were sent to a random sample of dentists and all OMFSs, aged 64 or younger with a known home and/or work address in the Netherlands. Contact details of these groups were made available by the Royal Dutch Dental Association (KNMT). This study is an extension of a previous study, in which only the outcomes of the dentists were described [6]. The questionnaire, which could be answered in writing or online (S1 File) contained 18 questions on different items and some general questions (i.e., gender, age, region, year, and country of graduation). The questions pertained to how often respondents received an extraction request without a clinical indication, how they dealt with this, and the patient's reasons for such a request.

In addition, a fictitious case was presented of a patient with dental phobia, reasonably good teeth, and a full extraction request. The case did not concern an actual patient but captured common elements of such cases based on the authors' years of experience in clinical practice. The answers to the fictitious case consisted of "did comply", "did not comply" or "did not know whether to comply". Before distributing the survey, it was piloted among five responding dentists who checked for completeness, correctness, and relevance. For this study, we distinguish two types of extractions without a valid clinical indication only. On the one hand, extractions are recommended by the dentist or the OMFS with a clear indication based on solid dental grounds. And on the other, extractions originate with the patient's request, whereas the dental professional lacks sound reasons grounded in dental science to recommend the extraction. We label the latter as extractions without a valid clinical indication.

### Data collection and procedure

The questionnaire was sent by mail in November 2019 by an independent research agency to 256 OMFSs and 800 dentists aged 64 years or younger. The participants could choose whether to fill in the paper version or an online version. With three to four weeks in between, a reminder was sent twice by letter. The collection of data was stopped at the end of January 2020. The research bureau anonymized the data and then forwarded the results to the principal author.

### Statistical analyses

Descriptive statistics were used to report the distribution and dispersion measures of the study variables. Differences in the distribution of percentages between OMFSs and dentists and by gender and age were analyzed using the Chi-Square test. A p-value of less than 0.05 was considered to be statistically significant. All analyses were performed using SPSS version 25.0.

### Ethical approval

The study concerned a request to dentists and OMFSs to answer a written or web-based questionnaire. In an accompanying letter, the dentists and OMFSs in the sample were explicitly informed that by submitting the answered questionnaire they consented to their data (in anonymized form) being used for analysis and reporting of the data collected through the web-based survey. These data do not relate to medical records or archived samples, but only to the

results of a questionnaire, which dentists and oral and maxillofacial surgeons were asked to answer without consulting any records or samples.

The study was conducted in accordance with the ethical requirements for research involving human subjects as outlined in the Declaration of Helsinki [17]. The study design was submitted to the Ethics Committee (METC) of the Academic Centre for Dentistry Amsterdam (ACTA). It concluded that this study is excepted from the regulations of the Dutch Medical Research Involving Human Subjects Act (nWMO, protocol number 201935).

## Results

The response rate was 28.1% (n = 72) for OMFSs and 30.3% (242) for dentists. Of these OMFSs, 27.8% were female and the mean age (SD) in this group was 47.0 (8.9). Given these characteristics, these OMFSs formed a representative reflection of the OMFS population in the Netherlands (female:19.0, p = 0.125 and mean age (SD) 46.2 (8.9), p = 0.500). Of the dentists in the study group, 48.3% were women and the mean age (SD) was 45.4 (11.8). In terms of gender, this group was representative of the population of dentists, but young dentists were slightly underrepresented (female: 53.0, p = 0.222 and mean age (SD) 42.2 (11.5): p = 0.001).

### Number of extraction requests on non-dental grounds

From January 2016 to November 2019, 81.9% (n = 59) of all OMFSs in our sample received a referral for a dental extraction of one or more teeth without a clear clinical indication. That was significantly more frequent than dentists (68.0%; n = 164) received in the same period (Chi Square = 5.224, df = 1, p = 0.022). For both OMFSs and dentists, gender and age showed no correlation with having received a non-dental extraction request.

### Reasons for a non-dental extraction request

When considering only the last non-dental extraction request, the most common reasons were financial and severe dental fear (Table 1). In 36.8% of the cases, the requests from the patients of OMFSs were financial only, 29.8% only of a psychological nature, 28.1% of both financial and psychological nature, and 5.3% of another nature. For dentists, the frequency of these

**Table 1. Frequency of extraction requests without any clinical indication as reported by OMFSs and dentists.**

| Reasons[a] | OMFS | | Dentist | |
|---|---|---|---|---|
| | n | % | N | % |
| Financial | 37 | 64.9 | 123 | 77.4 |
| Severe dental fear | 29 | 50.9 | 58 | 36.5 |
| Unexplained pain | 9 | 15.8 | 34 | 21.4 |
| Body Dysmorphic Disorder | | | 3 | 1.9 |
| No motivation to take care of teeth | 1 | 1.8 | 12 | 7.5 |
| Additional dental reason[b] | 2 | 3.5 | 4 | 2.5 |
| Cultural | | | 4 | 2.5 |
| Other[c] | 1 | 1.8 | 2 | 1.3 |
| N | 57 | | 159 | |

[a]Mentioning more than one reason was possible.

[b]Presence of a dental problem that in itself was not a reason for extraction.

[c]OMFS: medically compromised; dentist: unknown, aesthetics.

reasons differed somewhat: 49.7%, 18.2%, 27.7%, and 4.4% respectively, but this difference reached no statistical significance.

## Granting of a non-dental extraction request

Regarding the question as to how they had responded to the most recent patient who requested to extract a tooth while an apparent clinical reason was lacking, dentists agreed to such a request in significantly more cases than OMFSs (76.1% vs 61.4%; Chi Square = 4.999, df = 1, p = 0.034). It appeared that female OMFSs refused a non-dental extraction request much more often than their male colleagues (66.7% vs 25.6%; Chi Square = 8.746, df = 1, p = 0.003). There was no correlation with age. This also applied to the relationship between granting a non-dental extraction request and gender and age among dentists. While not statistically significant, OMFSs seemed to have hesitated less frequently than dentists before performing the extraction (26.3% vs 39.2%; Chi Square = 3.010, df = 1, p = 0.083). Only one of the OMFSs and a few dentists who performed an extraction without a valid clinical indication indicated they regretted their decision afterwards (2.9% vs 5.3%; Chi Square = 0.313, df = 1, p = 0.576).

## Regretting a non-dental extraction by patients

As far as the responding OMFSs knew, none of the patients whose last extraction was performed by them regretted this extraction, whereas 6.9% of the dentists reported that their patients had regrets. Conversely, OMFSs, more often than dentists, indicated that they did not know whether the patient regretted the extraction afterwards (64.7% vs 17.2%; Chi Square = 29.991, df = 2, p<0.001).

## Assessment of patients' mental competency

No difference was found between OMFSs and dentists when checking the decision-making competency of the patient who requested extraction in the absence of a clear clinical indication (82.5% vs 87.4%; Chi Square = 0.865, df = 1, p = 0.352). Table 2 shows how competence was determined. The most frequently cited reason for OMFSs not checking mental competency was that they did not doubt that the patient was mentally competent (60%). On the other hand, the most mentioned reason for dentists was that the patient did not live in an institution and could hence be presumed mentally competent (50%).

**Table 2. Methods used to assess a patient's intellectual competence concerning the most recent request for extraction without a clear clinical indication.**

| Way of checking[a] | OMFS | | Dentist | |
|---|---|---|---|---|
| | n | % | n | % |
| By asking (open) test questions to the patient | 42 | 89.4 | 125 | 91.9 |
| By asking the family/partner | 18 | 38.3 | 41 | 30.1 |
| In consultation with a colleague | 12 | 25.5 | 30 | 22.1 |
| In consultation with a physician | 4 | 8.5 | 8 | 5.9 |
| In consultation with a psychologist/psychiatrist | 2 | 4.3 | 2 | 1.5 |
| In another way, namely[b] | 2 | 4.3 | 7 | 5.1 |
| N | 47 | | 136 | |

[a]Mentioning more than one way of checking was possible.

[b]OMFS: consultation at a centre for special dental care, analysis by an orofacial pain specialist; Dentist: conversation guardian (3x), I wanted to consult with doctors and family but did not get permission from the patient, long process of months/years, no doubt about legal capacity, unknown.

## Fictitious case

The fictitious case we presented to OMFSs and dentists involved a 35-year-old patient with reasonably good teeth who requested total extraction because of a dental phobia. Virtually none of the practitioners indicated that they would perform the extraction (OMFS 5.7% vs dentists 3.0%, Chi Square = 3.805, df = 2, p = 0.149). In the group of OMFSs, this division of opinion showed no correlation with gender and age. This was the case in the group of dentists with regard to gender. Those who would perform the extraction were all male (5.7% vs 0.0%, Chi Square = 6.566, df = 1, p = 0.010). OMFS and dentists who said that they would grant the request in the fictitious case indicated, among other things, that they assumed that the patient had carefully and comprehensively considered or that he may decide for himself.

## Discussion

The present study, which is the first survey that examined reasons for extraction without a clinical indication, broken down into OMFSs and dentists in more detail, revealed that the most common reasons for extractions without a clinical indication were financial or related to the presence of severe dental fear. For OMFSs this appears in line with dentists. We are not able to compare our results with those from other studies because, to the best of our knowledge, there are no other studies that have addressed non-dental reasons for extraction [6].

A remarkable result is that, although the OMFSs were more likely to receive such an extraction request than dentists, dentists complied with a request for extraction more frequently than OMFSs. Interestingly, almost none of the dental professionals regretted complying with an extraction without a valid clinical indication afterwards. Another interesting result is that female OMFSs and dentists seem to be more reluctant to perform a non-dental extraction request than male OMFSs and dentists. Various studies show that female healthcare professionals differ from male healthcare professionals when it comes to patient communication: females are more focused on the patient [18]. It could be that female healthcare professionals enter into the conversation earlier and better with patients who want an extraction and, in more cases, can convince them not to perform the extraction. Whether this is the case requires further research.

To properly interpret the differences between OMFSs and dentists, it is important to understand that in the Netherlands, the working context of dentists and OMFSs is fundamentally different. Whereas dentists are likely to have a long-standing relationship with their patients and are familiar with their history, OMFSs are confronted with a new patient without much background information. The patient is frequently referred to the OMFS by their dentist "just" for carrying out the treatment with a referral letter consisting of a simple treatment order, not containing the treatment history, diagnosis, or indication for treatment. This context can easily result in different assessments and treatment options delivered by OMFSs compared to dentists. Moreover, it could be argued that their background and education make it more difficult for OMFSs than dentists to estimate how a tooth can be repaired, how to make it function again, and the extent to which extraction is needed. It should also be noted that, while the dentist almost always sets the primary indication for extraction, the OMFS bears the entire liability for the treatment. It is also possible that OMFSs receive referrals from general physicians without the intervention of a dentist. For example, patients who want an extraction (e.g., for financial reasons or dental fear) may be referred via this route because they do not have a dentist and do not want to go via a dentist. These differences between how dentists and OMFSs function within the Dutch health care system may partially explain why OMFSs received significantly more extraction requests without a clinical reason than dentists.

The present study shows that OMFSs were less likely than dentists to honour a request for a non-indicated. There may be various explanations for this reluctance. Firstly, as mentioned above, OMFSs typically do not have a long-standing relationship with the patient who presents to them with the request for extraction. Hence, OMFSs often lack a comprehensive understanding of the patient's particular needs, values, and circumstances to grant such an unusual request. Secondly, not having to continue treatment of the patient *after* the refusal (but instead referring the patient back to the primary dentist) may also render it easier to refuse. Thirdly, being the experts who typically have also to resolve complex side-effects of these extractions may cause OMFSs less eager to extract. And finally, there is the fact that Dutch OMFSs are all dual-trained as dentists and physicians and belong to the profession of dentistry and medicine. Their ethical decision-making is guided not only by the ethos and norms of the dental profession but also that of the medical profession. As is evident by stark differences between the codes of ethics of these two professions, with medical regulations often being more restrictive than dental codes of ethics, this too can contribute to the greater reluctance of OMFSs.

While dentists were shown to have granted more patient requests for extraction than OMFSs, there was no difference in how they responded to the fictitious case. This could mean that dentists are well aware of the ethics involved and the applicable legal rules and, in theory, are equally reluctant as OMFSs to grant these requests. However, clinical practice is a lot more 'unruly', and our research shows that it is not always easy to resist a patient who makes a 'demanding' request, even if one is not convinced that performing the extraction is ethically and legally the correct decision [19]. This may also explain why regrets about having a tooth extracted, even if the extraction has no valid dental reason, are uncommon within the dental profession. Only about three per cent of the OMFSs and about five per cent of the dentists admitted regretting their decision. That so few practitioners showed regret might be an example of "cognitive dissonance", a psychological concept explaining that if people experience inconsistencies in their thinking, they adjust their beliefs or behavior until they become consistent again [20]. As expected, many OMFS did not know whether their patients regretted their treatment because they usually do not see their patients again after the extraction. While none of the OMFSs indicated that their patients regretted the extraction, about seven per cent of the dentists reported that their patients expressed regrets about their most recent extraction. Of course, we should remind readers once more that these statistics about patients' regrets are based only on what the dental professionals reported. We did not survey the patients themselves. It may be that more patients regretted the extraction. But since they had requested the extraction, they may have been too embarrassed to admit it. It is also possible that some dental professionals "forgot" about some patients regretting the decision or did not "hear" the patients' regret. While such patient regrets could invoke a sense of "I was right in being so hesitant to grant the request", these regrets can also lead to legal troubles. Within a Dutch legal framework, dental professionals cannot hide behind a patient request [21]. They can be found guilty if they performed an intervention, even one explicitly requested by the patient, while they could not themselves, as healthcare professionals, support that intervention.

We found no difference between OMFSs and dentists in the frequency with which they assessed patients' decision-making competence. It seems that in most cases, neither group of dental professionals harbours doubts about the mental capacity of their patients and they, therefore, do not check this. Even if they also are trained as physicians and hence have extensive knowledge of mental healthcare issues that can impact a person's decision-making competence, OMFSs do not differ in this regard from dentists.

It is important to note that we have not found previous studies on the frequency with which OMFSs and dentists verify patient competence, so we do not know whether our findings are unusual. However, one would expect dental professionals to be *more* likely to question patients' competence when patients request interventions that pose ethical challenges and

even legal risks for the provider. But as mentioned, even when faced with a request for a non-indicated extraction that can have harmful severe side effects, most oral healthcare providers do not engage in a detailed and professional assessment of their patient's competency.

The present study has several limitations that need to be noted. Although a high response is always better because the chance of representativeness is greater, a response rate of around 30% is not particularly high. Conversely, such a response rate is considered satisfactory for survey research [22]. Further, a lower response does not necessarily mean that the results are not representative. To this end, the data collected in terms of gender and age are representative of the Dutch OMFS and dentist population. Although among the dentists the young dentists were slightly under-represented, the outcomes showed no correlation with age. Another limitation is that it is sometimes difficult to draw a rigid boundary between reasons for extraction with and without a clear clinical indication. For example, if a tooth can be preserved with endodontic treatment and a patient asks for extraction for a financial reason, extraction seems to be a treatment option that is within the standard of care. However, if a tooth in the view of the dental professional can quickly be restored, but a patient asks for extraction, there is no valid clinical indication. As described earlier, this is even more difficult for an OMFS to assess than for a dentist. Further complicating the situation is that the standard of care is not a hard dividing line. It is often difficult to define what the standard of care is. The standard is supposed to capture the entirety of professional medical insights that describe "proper action" in a particular situation. It hence reflects the shared values and norms within a professional group [23]. But when these values and norms vary considerably among the members of that group, the standard also becomes less sharply defined. Another limitation of this study is the fact that it may have been difficult for OMFS and dentists to report information relating to three years. Furthermore, the description of the fictitious case was also relatively brief, making it open to different interpretations. And perhaps the results would have been different if the patient in the case description had asked for an extraction of one or a few teeth rather than total clearance. Moreover, the fact that the study was conducted in the Netherlands means that, due to the differences in healthcare systems and the specific laws, regulations, and training that may differ per country, the results may not simply be extrapolated to other countries, affecting the generalizability of the results. In the Netherlands, an OMFS works on referral from a dentist. It would be interesting to carry out this study in a country where patients have direct access to the OMFS. Finally, the differences found between dentists and OMFSs, as well as the instances in which differences were expected but not found, evoke complex normative questions. We have sought to interpret these findings using prevailing ethical and legal principles. But more empirical research will be needed to explore exactly what reasons, emotions, ethical sensibilities, and/or legal concerns guide the decision-making of oral healthcare professionals when facing such challenging cases.

In conclusion, the current study results indicate that more than eight out of 10 OMFSs and almost seven out of 10 dentists regularly receive an extraction request while the clinical indication is unclear, mainly for financial reasons (OMFSs 65%; dentists 77%). Apparently, dentists are more likely to grant such a request and carry out the extraction. Mental competency was checked by almost all OMFSs (83%) and dentists (87%). Although few patients and dental professionals regret their decision afterwards, more research is required to determine possible long-term regrets and the reasons for these. These are important issues for future research because extraction is one of the most invasive dental procedures.

## Supporting information

**S1 File. Questionnaire "Non-dental reasons for tooth extraction".**
(DOCX)

## Author Contributions

**Conceptualization:** Dyonne Liesbeth Maria Broers, Leander Dubois, Jan de Lange, Jos Victor Marie Welie, Wolter Gerrit Brands, Maria Barbara Diana Lagas, Jan Joseph Mathieu Bruers, Ad de Jongh.

**Data curation:** Dyonne Liesbeth Maria Broers, Leander Dubois, Jan de Lange, Jos Victor Marie Welie, Wolter Gerrit Brands, Maria Barbara Diana Lagas, Jan Joseph Mathieu Bruers, Ad de Jongh.

**Formal analysis:** Dyonne Liesbeth Maria Broers, Leander Dubois, Jan de Lange, Jos Victor Marie Welie, Wolter Gerrit Brands, Maria Barbara Diana Lagas, Jan Joseph Mathieu Bruers, Ad de Jongh.

**Investigation:** Dyonne Liesbeth Maria Broers, Leander Dubois, Jan de Lange, Jos Victor Marie Welie, Wolter Gerrit Brands, Jan Joseph Mathieu Bruers, Ad de Jongh.

**Methodology:** Dyonne Liesbeth Maria Broers, Leander Dubois, Jan de Lange, Jos Victor Marie Welie, Wolter Gerrit Brands, Jan Joseph Mathieu Bruers, Ad de Jongh.

**Project administration:** Dyonne Liesbeth Maria Broers, Jan Joseph Mathieu Bruers, Ad de Jongh.

**Resources:** Dyonne Liesbeth Maria Broers, Jan Joseph Mathieu Bruers.

**Software:** Dyonne Liesbeth Maria Broers, Jan Joseph Mathieu Bruers, Ad de Jongh.

**Supervision:** Leander Dubois, Jan de Lange, Jos Victor Marie Welie, Wolter Gerrit Brands, Jan Joseph Mathieu Bruers, Ad de Jongh.

**Validation:** Dyonne Liesbeth Maria Broers, Leander Dubois, Jan de Lange, Jos Victor Marie Welie, Wolter Gerrit Brands, Jan Joseph Mathieu Bruers, Ad de Jongh.

**Visualization:** Dyonne Liesbeth Maria Broers, Leander Dubois, Jan de Lange, Jos Victor Marie Welie, Wolter Gerrit Brands, Jan Joseph Mathieu Bruers, Ad de Jongh.

**Writing – original draft:** Dyonne Liesbeth Maria Broers, Leander Dubois, Jan de Lange, Jos Victor Marie Welie, Wolter Gerrit Brands, Maria Barbara Diana Lagas, Jan Joseph Mathieu Bruers, Ad de Jongh.

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
