## [Decision Letter · Decision Letter 0]

1 Nov 2022

PONE-D-22-25258How dentists and oral and maxillofacial surgeons deal with tooth removal without a valid clinical indicationPLOS ONE

Dear Dr. Broers,

Thank you for submitting your manuscript to PLOS ONE. After careful consideration, we feel that it has merit but does not fully meet PLOS ONE’s publication criteria as it currently stands. Therefore, we invite you to submit a revised version of the manuscript that addresses the points raised during the review process.

ACADEMIC EDITOR: After careful reviewing of your manuscript, reviewers recommended a major revsion . Kindly, revise your manuscript according to the reviewer's comment's and suggestions />==============================

We look forward to receiving your revised manuscript.

Kind regards,

Essam Al-Moraissi

Academic Editor

PLOS ONE

Journal Requirements:

Additional Editor Comments:

Dear Authors,

After careful reviewing of your manuscript, reviewers recommended a major revsion . Kindly, revise your manuscript according to the reviewer's comment's and suggestions

Reviewers' comments:

Reviewer's Responses to Questions

**Comments to the Author**

1. Is the manuscript technically sound, and do the data support the conclusions?

Reviewer #1: Partly

Reviewer #2: No

Reviewer #3: Yes

2. Has the statistical analysis been performed appropriately and rigorously? 

Reviewer #1: Yes

Reviewer #2: No

Reviewer #3: Yes

3. Have the authors made all data underlying the findings in their manuscript fully available?

Reviewer #1: Yes

Reviewer #2: No

Reviewer #3: No

4. Is the manuscript presented in an intelligible fashion and written in standard English?

Reviewer #1: Yes

Reviewer #2: No

Reviewer #3: Yes

5. Review Comments to the Author

Reviewer #1: The paper requires the following revisions:

Language revision: the authors used the word “removal” in the title and text, it would be better changed to “extraction”. Also in Statistical analyses section, the sentence “The mean and the standard deviation SPSS version 25.0 were used to analyze the frequencies” should be revised

Introduction Section: the following sentences are more appropriate in the methods section and not in the introduction.

“But for this study, we distinguish two types only. On the one hand, extractions are recommended by the dentist or the OMFS with a clear indication based upon solid dental grounds. And on the other, extractions originate with the patient's request, whereas the dental professional lacks sound reasons grounded in dental science to recommend the extraction. We label the latter as extractions without a valid clinical indication”.

In the Material and Methods section: the Fictitious case is unrealistic (? Total clearance of sound teeth in a 35 years old) and should be a clinical scenario of a patient who wants extraction 1-2 teeth. Therefore, should be added as a limitation of the study.

Discussion section: needs revision, only 2 references were quoted. The authors should compare their results and justify their statements with previous studies (the authors reported many references in the introduction section nrs 6-16).

References: the authors should follow the journal style and make sure the references accuracy in text and reference list.

Reviewer #2: I read the article 'How dentists and oral and maxillofacial surgeons deal with tooth removal without a

valid clinical indication' with great interest, however, there are major flaws -

1. Simple questionnaire based study

2. Questions not validated

3. Poor response rate

4. Clinically not relevant

5. Title and short title are same [How dentists and oral and maxillofacial surgeons deal with tooth removal without a

valid clinical indication]

Reviewer #3: October 30, 2022

To: Dr. Al-Moraissi

Academic Editor

PLOS ONE

Dear Dr. Al-Moraissi

I write to you concerning the revision of the manuscript PONE-D-22-25258, titled “How dentists and oral and maxillofacial surgeons deal with tooth removal without a valid clinical indication”.

The study aimed to determine differences between oral and maxillofacial surgeons and dentists handling dental extractions without an evident clinical indication. The authors describe the study as a secondary analysis from a database of the Royal Dutch Dental Association (KNMT) (Broers et al., 2022). In general, the manuscript is well-written, provides interesting findings and seems suitable for publication in the PLOS ONE journal. However, some points need to be revised and/or clarified. I based my recommendations on the STROBE checklist.

Below, the point-to-point revision is presented.

Abstract

Comment 1: The “Study Design” section of the abstract is too short. The authors do not report important information such as the study design; the description of the setting (including dates); the eligibility criteria; the primary outcome; or the statistical methods. I understand the abstract has a limitation of words, but I believe the authors can give a detailed description.

Comment 2: Line 38: I suggest including both the percentages and the absolute values to clearly report the number of the recruited sample.

Comment 3: Line 44: "As for the request itself, it was found that 17.5% (n=47) of the OMFSs and 12.5% (n =140) of the dentists did not check for patients’ mental competency (p= 0.352)." I believe that the percentages or the absolute numbers are incorrect, please revise it.

Comment 4: In my opinion, the conclusion does not respond to the objective of the study. The authors should revise the text to strictly answer the objective.

Introduction

Comment 5: Line 77: The sentence “An extraction request without a clinical reason can be related to financial, cultural, or cultural issues” is confusing. Please, revise it.

Materials and Methods

Comment 6: The study design is not stated.

Comment 7: The authors should describe the eligibility criteria. Was any restriction applied to select the sample?

Comment 8: Was the questionnaire sent at the same period to both OMFSs and general dentists? I ask that because the OMFSs are not mentioned in the first published paper (Broers et al., 2022).

Comment 9: Line 137: “The mean and the standard deviation SPSS version 25.0 were used to analyze the frequencies.” I am not sure what the authors meant.

Comment 10: In the “Statistical analyses” section, I believe it is important that the authors clarify which dependent/independent variables were used in the statistical analyses.

Results

Comment 11: Lines 156-157: “The response rate was 28.1% for OMFSs (n=72; female 27.8%; mean age 47.0, SD 8.9) and 30.3% for dentists (n=242; female 48.3%; mean age 45.3, SD 11.8).” I am not sure if the mean ages are from the whole sample or only the female OMFSs/dentists.

Comment 12: Please, when describing the results, include both the percentages and the absolute values.

Comment 13: I believe it would be interesting to describe and explore the characteristics of the sample (such as sex, age, and years of graduation) in order to verify if these variables influenced the choice of extracting the teeth (or not).

Comment 14: In the tables (both Table 1 and 2), I am not sure how the percentages were calculated as they do not add up to 100%.

Discussion:

Comment 15: In the first paragraph of the discussion, I believe that the second sentence should be emphasized, once it responds to the main objective of the study.

References

Comment 16: The authors should revise the list of references, some of them (numbers 6 and 7) are incomplete.

6. PLOS authors have the option to publish the peer review history of their article (what does this mean?). If published, this will include your full peer review and any attached files.

Reviewer #1: **Yes: **Mawlood Kowash

Reviewer #2: No

Reviewer #3: No

---

## [Author Response · Author response to Decision Letter 0]

5 Dec 2022

To the Academic Editor of PLOS ONE

Dr. Al-Moraissi

Amsterdam, December 5th, 2022

Dear Dr. Al-Moraissi,

We would like to thank you and the reviewers for the review of our manuscript entitled: ”How dentists and oral and maxillofacial surgeons deal with tooth removal without a valid clinical indication”. We addressed the reviewer’s comments in this cover letter and the manuscript. The changes are highlighted in the new version of our manuscript.

Below you can find a point-by-point response to the comments.

Editor and Reviewer Comments to Author:

We carefully checked the manuscript and changed the format where needed.

Please provide additional details regarding participant consent. In the ethics statement in the Methods and online submission information, please ensure that you have specified what type you obtained (for instance, written or verbal, and if verbal, how it was documented and witnessed). If your study included minors, state whether you obtained consent from parents or guardians. If the need for consent was waived by the ethics committee, please include this information.

We changed it to: ‘In an accompanying letter, the dentists and OMFSs in the sample were explicitly informed that by submitting the answered questionnaire they consented to their data (in anonymized form) being used for analysis and reporting of the data collected through the web-based survey.’ [line 159 - 162] 

We added the same text to the Ethics Statement field of the submission form.

We note that you have indicated that data from this study are available upon request. PLOS only allows data to be available upon request if there are legal or ethical restrictions on sharing data publicly. For more information on unacceptable data access restrictions, please see http://journals.plos.org/plosone/s/data-availability#loc-unacceptable-data-access-restrictions. 

We added the sentence: “The coded data supporting the findings of this study are available and can be requested from the KNMT, via the Research Department (staatvandemondzorg@knmt.nl).’ [line ..-..] 

We added the sentences: “The data collection was commissioned by the Royal Dutch Dental Association (KNMT) and carried out by an independent research agency ('Third Party Research Institute'). It was determined that the KNMT remained the owner of the collected coded data. The coded data supporting the findings of this study are available and can be requested from the KNMT, via the Research Department (staatvandemondzorg@knmt.nl). Furthermore, it can be confirmed that the authors did not receive any special privileges in accessing the data that other researchers would not have.” [line 59 - 65]

The same method is used in the article of N. Bots e.a. (https://journals.plos.org/plosone/article?id=10.1371/journal.pone.0257561)

ACTA is the joint dental faculties of the Vrije Universiteit Amsterdam and the University of Amsterdam. There is an open-access deal with these faculties. Please use the Vrije Universiteit Amsterdam. A direct colleague of mine, dr N. Bots, published an article in PLOS One without payment. 

We included the caption for the Supporting Information file and have matched the in-text citation accordingly.

Reviewers' comments:

5. Review Comments to the Author

Reviewer #1: The paper requires the following revisions:

Language revision: the authors used the word “removal” in the title and text, it would be better changed to “extraction”. Also in Statistical analyses section, the sentence “The mean and the standard deviation SPSS version 25.0 were used to analyze the frequencies” should be revised

Thank you for this suggestion. We changed the word “removal” to “extraction”.

Besides, we added the sentence: “Descriptive statistics were used to report the distribution and dispersion measures of the study variables.” and: “All analyses were performed using SPSS version 25.0.” [line 151 – 152; line 154 - 155]

Introduction Section: the following sentences are more appropriate in the methods section and not in the introduction.

“But for this study, we distinguish two types only. On the one hand, extractions are recommended by the dentist or the OMFS with a clear indication based upon solid dental grounds. And on the other, extractions originate with the patient's request, whereas the dental professional lacks sound reasons grounded in dental science to recommend the extraction. We label the latter as extractions without a valid clinical indication”.

We agree with the reviewer. We moved these sentences to the methods section as requested. [line 135 - 140]

In the Material and Methods section: the Fictitious case is unrealistic (? Total clearance of sound teeth in a 35 years old) and should be a clinical scenario of a patient who wants extraction 1-2 teeth. Therefore, should be added as a limitation of the study. 

The fictitious case was about the total clearance of reasonably good teeth for a patient with dental phobia. In the Netherlands, with a history of high percentages of edentulous people, this is a realistic scenario. Maybe we could have added a fictitious case about extracting just one or a few elements as that may be more common. That’s why we added the phrase “And perhaps the results would have been different if the patient in the case description had asked for an extraction of one or a few teeth rather than total clearance” to the discussion. [line 368 - 370] 

Discussion section: needs revision, only 2 references were quoted. The authors should compare their results and justify their statements with previous studies (the authors reported many references in the introduction section nrs 6-16).

We’d like to thank the reviewer for this comment. We added some sentences and six references: “For OMFSs this appears in line with dentists.” [line 256] and “We are not able to compare our results with those from other studies because, to the best of our knowledge, there are no other studies that have addressed non-dental reasons for extraction” [line 256- 259] and “ Although a high response is always better because the chance of representativeness is greater, a response rate of around 30% is not particularly high. Conversely, such a response rate is considered satisfactory for survey research.” [line 347 - 350] and another three references [line 267; line 312; line 330] 

References: the authors should follow the journal style and make sure the references accuracy in text and reference list. 

Thank you for this comment. We checked the journal style and changed the references accordingly, if necessary.

Reviewer #2: 

I read the article 'How dentists and oral and maxillofacial surgeons deal with tooth removal without a valid clinical indication' with great interest, however, there are major flaws -

1. Simple questionnaire based study

Thank you for this comment. It is indeed a survey, a questionnaire-based study. A very common way of mapping beliefs and explanations about behaviour and experiences. As such, it has already generated a great deal of useful knowledge and formed the impetus for follow-up research. So, we do not agree with the somewhat negative connotation.

2. Questions not validated

Indeed, the questions have not been validated. But that is difficult in the first place if the topic has never been questioned before. In addition, most questions pertain to actual behaviour, making validation less important. Validation is particularly important for questions where questions are asked about 'latent' concepts. For example, when mapping 'fear', 'happiness', etc., validation is important.

3. Poor response rate

We’d like to thank the reviewer for this comment. A high response is always better because the chance of representativeness is greater. However, a response rate of around 30% is not particularly high, but satisfactory for survey research. 

Meng-Jia Wu, Kelly Zhao, Francisca Fils-Aime. Response rates of online surveys in published research: A meta-analysis. Computers in Human Behavior Reports, Volume 7, 2022, 100206, ISSN 2451-9588, https://doi.org/10.1016/j.chbr.2022.100206.

But a lower response does not necessarily mean that the results are not representative. That may very well be the case, albeit with slightly larger confidence intervals. In this study, the data collected in terms of gender and age are representative of the OMFS and dentist population in the Netherlands. Only among the dentists was a slight under-representation of young dentists. Incidentally, the outcomes showed no correlation with age, so this slight under-representation was acceptable.

We added the sentences: “Although a high response is always better because the chance of representativeness is greater, a response rate of around 30% is not particularly high. Conversely, such a response rate is considered satisfactory for survey research. Further, a lower response does not necessarily mean that the results are not representative. To this end, the data collected in terms of gender and age are representative of the Dutch OMFS and dentist population. Although among the dentists the young dentists were slightly under-representated, the outcomes showed no correlation with age.” [line 347 - 354]

4. Clinically not relevant

We think the outcomes are clinically relevant because extractions on non-dental grounds cause unnecessary and irreversible damage and it is, therefore, important to investigate how often this occurs and what practitioners consider whether to accept such an extraction request.

5. Title and short title are same [How dentists and oral and maxillofacial surgeons deal with tooth removal without a

valid clinical indication]

We would like to thank the reviewer for this comment. We changed the title to: ”How dentists and oral and maxillofacial surgeons deal with tooth extraction without a valid clinical indication” and the short title to “Tooth extraction without a clinical indication” [line 1 - 2]

Reviewer #3

The study aimed to determine differences between oral and maxillofacial surgeons and dentists handling dental extractions without an evident clinical indication. The authors describe the study as a secondary analysis from a database of the Royal Dutch Dental Association (KNMT) (Broers et al., 2022). In general, the manuscript is well-written, provides interesting findings and seems suitable for publication in the PLOS ONE journal. However, some points need to be revised and/or clarified. I based my recommendations on the STROBE checklist.

Below, the point-to-point revision is presented.

Abstract

Comment 1: The “Study Design” section of the abstract is too short. The authors do not report important information such as the study design; the description of the setting (including dates); the eligibility criteria; the primary outcome; or the statistical methods. I understand the abstract has a limitation of words, but I believe the authors can give a detailed description.

We agree with the reviewer. We added the sentence: “Respondents could answer the questions in writing or online. The data was collected in the period from November 2019 to January 2020, during which two reminders were sent. Analysis of the data took place via descriptive statistics and Chi Square test.” [line 36 - 39]

The primary outcome is expressed in the objective.

Comment 2: Line 38: I suggest including both the percentages and the absolute values to clearly report the number of the recruited sample.

The reviewer is right. We extended this sentence to: ’The response rate was 28.1% (n=72) for OMFSs and 30.3% (n=242) for dentists.’ [line 41]

Comment 3: Line 44: "As for the request itself, it was found that 17.5% (n=47) of the OMFSs and 12.5% (n =140) of the dentists did not check for patients’ mental competency (p= 0.352)." I believe that the percentages or the absolute numbers are incorrect, please revise it.

Thank you for this comment. This is an omission. We changed the sentence to: ‘As for the request itself, it was found that 17.5% (n=10) of the OMFSs and 12.5% (n =20) of the dentists did not check for patients’ mental competency (p= 0.352).’ [line 47 - 48]

Comment 4: In my opinion, the conclusion does not respond to the objective of the study. The authors should revise the text to strictly answer the objective.

Thank you for this comment. We agree with the reviewer. We added the lacking information to the conclusion: ‘In conclusion, the current study results indicate that more than eight out of 10 OMFSs and almost seven out of 10 dentists regularly receive an extraction request while the clinical indication is unclear, mainly for financial reasons (OMFSs 65%; dentists 77%). Apparently, dentists are more likely to grant such a request and carry out the extraction. Mental competency was checked by almost all OMFSs (83%) and dentists (87%)’ [line 381 - 385]

Introduction

Comment 5: Line 77: The sentence “An extraction request without a clinical reason can be related to financial, cultural, or cultural issues” is confusing. Please, revise it.

We agree with the reviewer. We changed this sentence to “An extraction request without a clinical reason can be related to financial, or cultural issues” [line 82 - 83]

Materials and Methods

Comment 6: The study design is not stated.

We added some sentences: “For this survey, questionnaires were sent to a random sample of dentists and all OMFSs, aged 64 or younger with a known home and/or work address in the Netherlands. Contact details of these groups were made available by the Royal Association of Dentistry (KNMT). This study is an extension of a previous study, in which only the outcomes of the dentists were described.(6) The questionnaire, which could be answered in writing or online.” [line 121 - 125]

Comment 7: The authors should describe the eligibility criteria. Was any restriction applied to select the sample?

 Thank you for this comment. We added the sentences as mentioned in comment 6.

Comment 8: Was the questionnaire sent at the same period to both OMFSs and general dentists? I ask that because the OMFSs are not mentioned in the first published paper (Broers et al., 2022).

Yes, the questionnaire was sent at the same period to both OMFSs and general dentists.

Comment 9: Line 137: “The mean and the standard deviation SPSS version 25.0 were used to analyze the frequencies.” I am not sure what the authors meant.

We’d like to thank the reviewer for this comment. We changed this sentence to: “Descriptive statistics were used to report the distribution and dispersion measures of the study variables.” [line 151 - 152]

Comment 10: In the “Statistical analyses” section, I believe it is important that the authors clarify which dependent/independent variables were used in the statistical analyses.

Dependent variables are having dealt with a request for non-dental extraction and whether or not such a request has been honoured. The independent variables are dental profession, gender and age. 

We think that is clear from the questions what the dependent variables are. Regarding the independent variables we changed the sentence to “Differences in the distribution of percentages between OMFSs and dentists and by gender and age were analyzed using the Chi-Square test.” [line 152 - 153]

Results

Comment 11: Lines 156-157: “The response rate was 28.1% for OMFSs (n=72; female 27.8%; mean age 47.0, SD 8.9) and 30.3% for dentists (n=242; female 48.3%; mean age 45.3, SD 11.8).” I am not sure if the mean ages are from the whole sample or only the female OMFSs/dentists.

The reviewer is right. We extended this sentence to: ‘The response rate was 28.1% (n=72) for OMFSs and 30.3% (242) for dentists. Of these OMFSs, 27.8% were female and the mean age (SD) in this group was 47.0 (8.9). Given these characteristics, these OMFSs formed a representative reflection of the OMFS population in the Netherlands (female:19.0; p=0.125 and mean age (SD) 46.2 (8.9): p=0.500). Of the dentists in the study group, 48.3% were women and the mean age (SD) was 45.4 (11.8). In terms of gender, this group was representative of the population of dentists, but young dentists were slightly underrepresented (female: 53.0; p=0.222 and mean age (SD) 42.2 (11.5): p=0.001).’ [line 172 - 179] 

Comment 12: Please, when describing the results, include both the percentages and the absolute values.

We thank the reviewer for this comment. The absolute numbers are shown in the tables. It is a lot of duplicate information to also include the absolute numbers in the text. We don’t think that improves readability either.

Comment 13: I believe it would be interesting to describe and explore the characteristics of the sample (such as sex, age, and years of graduation) in order to verify if these variables influenced the choice of extracting the teeth (or not).

We fully agree with this comment. We added the sentences: “For both OMFSs and dentists, gender and age showed no correlation with having received a non-dental extraction request.” [line 185 - 186] and “It appeared that female OMFSs refused a non-dental extraction request much more often than their male colleagues (66.7% vs 25.6%; Chi Square=8.746, df=1, p=0.003). There was no correlation with age. This also applied to the relationship between granting of a non-dental extraction request and gender and age among dentists.” [line 206 - 209] and “In the group of OMFSs, this division of opinion showed no correlation with gender and age. This was the case in the group of dentists with regard to gender. Those who would perform the extraction were all male (5.7% vs 0.0%, Chi Square=6.566, df=1, p=0.010).” [line 244 - 247] and “Another interesting result is that female OMFSs and dentists seem to be more reluctant to perform a non-dental extraction request than male OMFSs and dentists. This study and others show that female healthcare professionals differ from male healthcare professionals when it comes to patient communication: females are more focused on the patient. It could be that female healthcare professionals enter into the conversation earlier and better with patients who want an extraction and, in more cases, can convince them not to perform the extraction. Whether this is the case requires further research.” [line 263 - 270]

Comment 14: In the tables (both Table 1 and 2), I am not sure how the percentages were calculated as they do not add up to 100%.

The percentages in both tables relate to the N-values stated in both tables. But because in table 1 OMFSs and dentists could indicate more than one reason (i.e. combination of reasons), the percentages add up to more than 100. The same applies to table 2, for ‘way of checking’. This is indicated in both tables with a footnote.

Discussion:

Comment 15: In the first paragraph of the discussion, I believe that the second sentence should be emphasized, once it responds to the main objective of the study.

We thank the reviewer for this remark. We agree with the reviewer. We changed the sentence “The study also showed that although the OMFSs were more likely to receive such an extraction request than dentists, dentists complied with a request for extraction more frequently than OMFSs” to “A remarkable result is that although the OMFSs were more likely to receive such an extraction request than dentists, dentists complied with a request for extraction more frequently than OMFSs.” [line 260 - 262]

References

Comment 16: The authors should revise the list of references, some of them (numbers 6 and 7) are incomplete.

We would like to thank the reviewer for this comment. We added the missing information to the reference list. [line 402 - 407]

---

## [Decision Letter · Decision Letter 1]

26 Dec 2022

How dentists and oral and maxillofacial surgeons deal with tooth extraction without a valid clinical indication

PONE-D-22-25258R1

Dear Dr. Broers,

We’re pleased to inform you that your manuscript has been judged scientifically suitable for publication and will be formally accepted for publication once it meets all outstanding technical requirements.

Kind regards,

Essam Al-Moraissi

Academic Editor

PLOS ONE

Additional Editor Comments (optional):

Reviewers' comments:

Reviewer's Responses to Questions

**Comments to the Author**

1. If the authors have adequately addressed your comments raised in a previous round of review and you feel that this manuscript is now acceptable for publication, you may indicate that here to bypass the “Comments to the Author” section, enter your conflict of interest statement in the “Confidential to Editor” section, and submit your "Accept" recommendation.

Reviewer #1: All comments have been addressed

2. Is the manuscript technically sound, and do the data support the conclusions?

Reviewer #1: Yes

3. Has the statistical analysis been performed appropriately and rigorously? 

Reviewer #1: Yes

4. Have the authors made all data underlying the findings in their manuscript fully available?

Reviewer #1: Yes

5. Is the manuscript presented in an intelligible fashion and written in standard English?

Reviewer #1: Yes

6. Review Comments to the Author

Reviewer #1: Apparently the authors have addressed my all comments and satisfactorily made the necessary changes.

7. PLOS authors have the option to publish the peer review history of their article (what does this mean?). If published, this will include your full peer review and any attached files.

Reviewer #1: **Yes: **Mawlood Kowash

---

## [Editor Report · Acceptance letter]

5 Jan 2023

PONE-D-22-25258R1 

How dentists and oral and maxillofacial surgeons deal with tooth extraction without a valid clinical indication 

Dear Dr. Broers:

I'm pleased to inform you that your manuscript has been deemed suitable for publication in PLOS ONE. Congratulations! Your manuscript is now with our production department. 

Kind regards, 

on behalf of

Dr. Essam Al-Moraissi 

Academic Editor

PLOS ONE